# Circulating Omega-3 Polyunsaturated Fatty Acids Levels in Coronary Heart Disease: Pooled Analysis of 36 Observational Studies

**DOI:** 10.3390/nu16111610

**Published:** 2024-05-24

**Authors:** Yanan Xiao, Yifang Chen, Anne Pietzner, Ulf Elbelt, Zhimin Fan, Karsten H. Weylandt

**Affiliations:** 1Department of Medicine, Pingxiang People’s Hospital, Gannan Medical University, Pingxiang 337000, China; yanan.xiao@mhb-fontane.de; 2Division of Medicine, Department of Gastroenterology, Metabolism and Oncology, University Hospital Ruppin-Brandenburg, Brandenburg Medical School, 16816 Neuruppin, Germany; 3Faculty of Health Sciences, Joint Faculty of the Brandenburg University of Technology, Brandenburg Medical School and University of Potsdam, 14467 Potsdam, Germany; 4Medical Department, Division of Psychosomatic Medicine, Campus Benjamin Franklin, Charité-Universitätsmedizin Berlin, Corporate Member of Freie Universität Berlin and Humboldt-Universität zu Berlin, 10117 Berlin, Germany; 5Department of Cardiology, Beijing Anzhen Hospital, Beijing Institute of Heart, Lung and Blood Vessel Diseases, Capital Medical University, Beijing 100029, China

**Keywords:** *n*-3 polyunsaturated fatty acid, coronary heart disease, cardiovascular disease, eicosapentaenoic acid, docosapentaenoic acid

## Abstract

Long-chain *n*-3 polyunsaturated fatty acid (PUFA) supplementation has shown potential benefits in the prevention of coronary heart disease (CHD); however, the impact of omega-3 fatty acid levels on CHD risk remains a subject of debate. Here, we aimed to investigate the association between *n*-3 PUFA levels and the risk of CHD, with particular reference to the subtypes of *n*-3 PUFA. Methods: Prospective studies and retrospective case-control studies analyzing *n*-3 PUFA levels in CHD, published up to 30 July 2022, were selected. A random effects meta-analysis was used for pooled assessment, with relative risks (RRs) expressed as 95% confidence intervals (CIs) and standardized mean differences expressed as weight mean differences (WMDs). Subgroup and meta-regression analyses were conducted to assess the impact of *n*-3 PUFA exposure interval on the CHD subtype variables of the study. Results: We included 20 prospective studies (cohort and nested case-control) and 16 retrospective case-control studies, in which *n*-3 PUFAs were measured. Higher levels of *n*-3 PUFAs (ALA, EPA, DPA, DHA, EPA + DHA, total *n*-3 PUFAs) were associated with a reduced risk of CHD, with RRs (95% CI) of 0.89 (0.81, 0.98), 0.83 (0.72, 0.96); 0.80 (0.67,0.95), 0.75 (0.64, 0.87), 0.83 (0.73, 0.95), and 0.80 (0.70, 0.93), respectively, *p* < 0.05. CHD patients had significantly lower *n*-3 PUFA levels compared to healthy controls (*p* < 0.05). In the subgroup analysis, a significant inverse trend was found for both fatal CHD and non-fatal CHD with *n*-3 PUFA (EPA + DHA) levels. Also, the link between *n*-3 PUFA levels in erythrocytes with total CHD was generally stronger than other lipid pools. Conclusions: *n*-3 PUFAs are significantly related to CHD risk, and these findings support the beneficial effects of *n*-3 PUFAs on CHD.

## 1. Introduction

The role of *n*-3 polyunsaturated fatty acids (PUFAs) in cardiovascular disease remains highly controversial. *n*-3 PUFAs, mainly α-linolenic acid (ALA), eicosapentaenoic acid (EPA), docosapentaenoic acid (DHA), and docosapentaenoic acid (DPA), were reported in the 1970s on a small group of Eskimos showing cardioprotective effects, which has ignited extensive interest and research into *n*-3 PUFAs within the scientific community [1]. In the following decades, scientists have provided a comprehensive explanation of the cardiovascular protective mechanisms of *n*-3 PUFA, and it is generally accepted to be effective in improving blood lipids and combating hypertension [2,3]. In addition, the specialized pro-resolving lipid mediators (SPMs) of EPA and DHA are also thought to mediate vascular anti-inflammation and oxidation [4,5]. As recommended by the AHA Dietary Guidelines since 2003, the health benefits of *n*-3 PUFAs for both primary and secondary prevention of cardiovascular disease can be achieved by eating at least two servings of fish per week [6].

Several large prospective cohort studies and randomized controlled trials have supported an inverse association between *n*-3 PUFA and the risk of cardiovascular disease [7,8,9,10]. However, many subsequent studies have not been entirely consistent [11,12,13,14]. A meta-analysis by Aung et al. of 10 trials with 77,917 participants revealed that supplementation with marine-derived *n*-3 PUFAs for a mean period of 4.4 years did not significantly lower major vascular events [15]. *n*-3 PUFA could decrease the risk of myocardial infarction (MI), total cardiovascular disease (CVD), and CVD death, according to Hu’s research utilizing a fixed-effects model [16]. A more recent and comprehensive meta-analysis conducted by Shen and colleagues revealed that supplementing with *n*-3 PUFA, regardless of dose, effectively prevented major adverse cardiovascular events (MACE), cardiovascular death, and MI, but not with all-cause mortality, stroke, or revascularization [17]. In contrast, in a pooled analysis of 79 randomized controlled trials (RCTs), Abdelhamid et al. found that *n*-3 PUFAs had only a minimal impact on MACE, whereas ALA could reduce coronary heart disease (CHD) mortality [18].

These conflicting results could be caused by several factors. Levels of *n*-3 PUFA can be influenced by genetics and dietary habits [19]. Additionally, certain medications can also impact the metabolism and overall concentration of these fatty acids (FAs) [20]. Another aspect to consider is the reliance of most studies on self-reported dietary intake of *n*-3 PUFA or fish oil. These types of reports are prone to inaccuracies in reflecting an individual’s *n*-3 PUFA levels and intake, which may weaken the reliability of findings [21]. In light of these challenges, utilizing objective measurements, such as biomarkers, can help to overcome the limitations of self-reported data. 

To our knowledge, only a few pooled analyses have been conducted to report the association of individual levels of *n*-3 PUFA with cardiovascular risk [22,23,24]. Of these, research concerning CHD primarily dated back to before 2016 and has not been updated in several years. Meanwhile, no pooled studies have yet specifically focused on comparing blood *n*-3 PUFA levels between patients with CHD and healthy controls. Given this, we have updated the latest study and strictly distinguished retrospective and prospective studies assessing different types of *n*-3 PUFAs (ALA, EPA, DPA, DHA) on CHD and CHD subgroup (total CHD, fatal and non-fatal CHD) risk factors. Finally, we assessed the differences in *n*-3 PUFA levels between patients with and without CHD.

## 2. Materials and Methods

The study adheres to PRISMA guidelines [25] and was registered at PROSPERO (CRD42022340248).

### 2.1. Search Strategy

PubMed, Embase, Web of Science, and Cochrane databases were comprehensively reviewed for English-language literature from their inception to July 2022 in humans of the association between *n*-3 PUFAs and CHD. The primary terms used were “*n*-3 PUFAs” and “CHD”. To enhance the breadth of the search, we included synonyms and related terms for omega-3 fatty acids, such as “*n*-3 fatty acids”, “marine oils”, “EPA”, “DHA”, and “ALA”. Similarly, for CHD, we incorporated related terms such as “CHD”, “ischemic heart disease”, “cardiovascular disease”, and specific conditions under the CHD umbrella, like “myocardial infarction” and “angina pectoris”. The detailed strategy is shown in Appendix A.

### 2.2. Quality Assessment

We assessed the quality of cohort studies using the Newcastle–Ottawa Scale (NOS). The NOS evaluates three aspects of non-randomized studies: (1) selection, (2) comparability, and (3) assessment of either exposure or outcomes of interest. Studies can receive 2 to 4 stars within each NOS domain, respectively (Appendix A).

### 2.3. Exposures and Outcomes

Assessment of FA levels: We extracted different types of *n*-3 PUFAs (ALA, EPA, DPA, DHA) separately and the studies included in our analysis measured FAs in at least one compartment, including whole blood, red blood cells, serum, plasma, or adipose tissue. Assessment of CHD: Total CHD events were defined as non-fatal CHD (angina, non-fatal myocardial infarction (MI)), and fatal CHD (fatal MI and sudden cardiac death). For the studies reporting biomarkers of *n*-3 PUFA and CHD risk, inclusion criteria were: (1) prospective cohort research (nested case-control); (2) at least one type of *n*-3 PUFA concentration was measured; (3) the primary outcomes were risk estimates (relative risks (RRs), odds ratios (ORs), and hazard ratios (HRs)) with 95% CI for CHD events. For studies comparing *n*-3 PUFA levels in patients with and without CHD, the inclusion criteria were: (1) adult CHD patients and matched controls; (2) retrospective case-control study design; (3) at least one type of *n*-3 PUFA levels measured between two groups. The included trials were approved by their respective institutional review boards, and we assessed the risk of bias in these trials.

### 2.4. Data Extraction

Two authors (Yanan Xiao and Yifang Chen) independently conducted study selection based on the criteria and extracted data as follows: (1) basic information (author, year, geographical area, study type/design, sample size, etc.); (2) participant characteristics; (3) different types of *n*-3 PUFAs (ALA, EPA, DPA, DHA, EPA + DHA); (4) lipid pools (whole blood, erythrocytes, serum, plasma); (5) mean of *n*-3 PUFA levels; (6) total CHD or CHD subgroup endpoint events; (7) multivariate-adjusted risk estimates with 95% confidence intervals (CI).

### 2.5. Data Processing and Statistical Analysis

We recorded RRs or HR estimates and their 95% CIs for prospective studies (cohort and nested case control). Adjusted HRs were utilized and treated as equivalent risk measures to RRs. When the odds ratio (OR) varies between 0.5 and 2.5, we consider the OR and RR to be approximately equal [26]. The pooled analyses were conducted on the log scale of the RRs. For data with different categories of tertile, quartile, quintile, or quintile, we recorded the relative highest and lowest category valuations. For the retrospective case-control studies, we calculated the weight mean difference (WMD) in tissue and circulation by combining the mean and standard deviation (SD) of data from different types of *n*-3 PUFAs (ALA, EPA, DPA, DHA) in the CHD patients and controls. In cases where studies reported EPA and DHA levels separately, we calculated the weighted mean difference (WMD) to combine the EPA and DHA levels to illustrate the difference in blood levels of *n*-3 PUFAs in those patients with and without CHD. Finally, the degree of heterogeneity between these trials was calculated using the Q test and the *I*^2^ statistic. A random-effects model was applied [27,28].

### 2.6. Subgroup Analysis

We conducted subgroup analyses to assess the impact of different subtypes on the outcomes, focusing on (1) the lipid pool of *n*-3 PUFA exposure (e.g., erythrocytes, serum, plasma, whole blood, adipose tissue); (2) specific CHD outcomes (e.g., total CHD, fatal and non-fatal CHD). If studies reported data for EPA and DHA separately, we pooled the RR values for EPA + DHA using a random effects model.

Meta-regression and subgroup stratification were employed to evaluate potential sources of heterogeneity between studies. Funnel plots and Egger’s test were used to explore the potential publication bias [29]. Data analyses were carried out with Stata (version 16.0), and *p* < 0.05 was deemed statistically significant.

## 3. Results

We reviewed 4761 papers identified through database searches and other sources. Thirty-seven studies ultimately met the inclusion criteria and were included in the review after screening titles and full-text contents. The detailed flow chart is shown in Appendix A.

The characteristics of the 20 prospective cohorts and 16 retrospective case-control studies are presented (Table 1 and Table 2). The quality of the included observational studies was assessed using the NOS. The NOS evaluation criteria and results are given in Appendix A. In Table 1, a total of 28,952 participants representing 15 national areas, with a mean age ranging from 46.4 to 79.1 years at baseline, and the proportion of enrolled women ranged from 0% to 100%. Each study quantified the levels of *n*-3 PUFAs in the circulation system or tissues, with 2 studies focusing on erythrocytes [10,30], 10 studies on plasma [31,32,33,34,35,36,37,38,39,40], 4 studies on serum [41,42,43,44], and 4 studies on whole blood [45,46,47,48]. In Table 2, 1148 CHD patients and 1156 healthy controls were enrolled in the study of compared *n*-3 PUFA levels. These participants were from 11 countries and ranged in age from 18 to 62.7 years, with five studies analyzing PUFA in erythrocytes [49,50,51,52,53], two studies in plasma [54,55], four studies in serum [56,57,58,59], one study in whole blood [60], and four studies for tissue or adipose tissue, respectively [61,62,63,64].

We examined the relationships between various types of *n*-3 PUFAs and the risk of CHD. Elevated levels of *n*-3 PUFAs (ALA, EPA, DPA, DHA, EPA + DHA) were linked with reduced CHD risk (Figure 1). Specifically, 13 studies reported the association between ALA and CHD events, with a pooled RR of 0.89 (0.81–0.98), *p* = 0.02, and heterogeneity: *p* = 0.12, *I*^2^ = 32.6%; 14 trials for EPA, RR: 0.83 (0.72–0.96), *p* < 0.01, heterogeneity: *p* < 0.01, *I*^2^ = 65.6%; 9 trials for DPA, RR: 0.80 (0.67–0.95), *p* < 0.01, heterogeneity: *p* < 0.01, *I*^2^= 62.0%; 14 trials reported DHA, RR: 0.75 (0.64–0.87), *p* < 0.01, heterogeneity: *p* < 0.01, *I*^2^ = 66.6%. In addition, 8 and 11 reported CHD events for EPA + DHA and the total *n*-3 PUFAs, resulting in a summary RR of 0.83 (0.73–0.95) and RR: 0.80 (0.70–0.93), *p* < 0.05, with heterogeneity: *I*^2^ = 67.4% and *I*^2^ = 66.6%, *p* < 0.01, respectively. The overall result between *n*-3 PUFA and CHD was 0.83 (95% CI: 0.79–0.87). In the CHD model of ALA, the heterogeneity between studies was low, and the other models were highly heterogeneous.

A total of 16 studies compared blood levels of *n*-3 PUFAs in patients with and without CHD (Figure 2), and overall, the levels of different types of *n*-3 PUFA (EPA, DHA, DPA, EPA + DHA, and total *n*-3 PUFAs) were inversely related to CHD, but not with ALA. Also, 13, 10, 14, and 5 studies provided complete data for EPA, DHA, DPA, EPA + DHA, and total *n*-3 PUFAs, the levels of each of these FAs were significantly lower in the CHD group (WMD = −0.33; 95% CI: −0.53 to −0.12; WMD = −0.27; 95% CI: −0.56 to 0.02; WMD = −0.44; 95% CI: −0.73 to −0.16, WMD = −0.63; 95% CI: −1.05 to −0.21; WMD = −0.61; 95% CI: −1.11 to −0.11; *p* < 0.01, respectively). In the 12 case-control studies, ALA levels did not change significantly in patients with and without CHD (WMD = −0.01, 95% CI: −0.30 to 0.28). Heterogeneity among studies comparing *n*-3 PUFA levels in patients with CHD versus those without CHD was substantial (*I*^2^: 77.9%–91.8%, *p* < 0.01).

We conducted a subgroup analysis by combining the RR estimates of EPA and DHA, which included a total of 20 studies (Figure 3, Appendix A). For the different subgroups of clinical endpoint, the observations were consistent for total CHD, fatal and non-fatal CHD (RR 0.81 (0.70–0.93 for total CHD; RR 0.74 (0.57–0.97) for fatal CHD; RR 0.74 (0.60–0.92) for non-fatal CHD, *p* < 0.05 for both) (Figure 3A). Regarding the different subgroups of EPA + DHA exposure compartment, whole blood, plasma, and serum had stronger correlations with CHD (RR 0.77 (0.71–0.85) for whole blood; RR 0.81 (0.70–0.93) for plasma; RR 0.79 (0.64–0.76) for serum, RR 0.43 (0.24–0.76) for erythrocytes, *p* <0.05 for both) (Figure 3B).

Funnel plot visualization and assessment via Egger’s test found no identified evidence of significant publication bias. Sensitivity analyses also confirmed the stability and reliability of the results produced by our statistical models. Additionally, meta-regression analysis indicated that the heterogeneity among the subgroups was not significant (Appendix A).

## 4. Discussion

The present study was a comprehensive analysis of individual-level data from 20 prospective cohort studies (cohort and nested case-control) and 16 independent case-control studies. Overall, our findings indicated that higher circulating levels of *n*-3 PUFAs (ALA, EPA, DPA, DHA, EPA + DHA) were related to a lower CHD risk. In subgroup analyses, EPA + DHA values exhibited a strong correlation with both fatal and non-fatal CHD, as well as total CHD. The association between *n*-3 PUFAs and CHD risk was generally more robust when assessed in erythrocyte lipid pools. In addition, the level of *n*-3 PUFAs (EPA, DPA, DHA) was significantly lower in patients with CHD compared to control subjects without CHD but with substantial heterogeneity. The ALA concentrations were similar in the two groups.

Our findings are generally in line with earlier evidence [22,23,24], but there were exceptions [15]. Gobbo et al.’s [23] pooled analysis of 19 prospective cohort studies reported that higher circulating levels of the omega-3 fatty acids ALA, DPA, and DHA were correlated with lower RRs of fatal CHD events, with RRs ranging from 0.90 to 0.91. Particularly, DPA was also associated with a reduced risk of total CHD; however, the results were not significant for EPA and CHD. Additionally, this study found generally stronger correlations for *n*-3 PUFA levels measured in phospholipids and total plasma versus other lipid fractions. A more recent meta-analysis by Harris et al. [22] has provided a broader perspective by evaluating the collective evidence, and their research indicated that DHA and EPA + DHA were associated with a 15% to 18% reduction in total mortality, especially CVD, whereas ALA was not.

In recent meta-analyses of RCTs, Bernasconi et al. [65] updated their dataset to include the two latest studies, further confirming that *n*-3 PUFAs still have favorable effects on CVD events. However, Rizos’s research [66] emphasized that the benefits were apparent only with high-dose administration. Additionally, Chao et al.’s [67] analysis revealed that protective effects were only significant in subgroups of intervention with EPA and baseline triglyceride (TG) ≥ 1.7 mmol/L, but they did not compare the blood *n*-3 PUFA levels between patients with CHD and controls.

The initial two large-scale randomized trials, GISSI-Prevenzione and JELIS [7,8], showed a 14–20% reduction in major coronary events in individuals with CHD using *n*-3 PUFA supplements of 1–1.8 g. Similar results were later observed in the REDUCE-IT trial [9], which administered a higher dose of 4 g of EPA to statin-treated patients with elevated TGs. In this study, the risk of total cardiovascular death, non-fatal myocardial infarction, and non-fatal stroke was reduced by over 20% compared to the control group. However, the outcomes of other RCTs using only 840 mg of EPA + DHA were less favorable. ORIGIN and ASCEND found that *n*-3 PUFAs decreased TGs while having no impact on cardiovascular events in patients with type 2 diabetes [12,68]. The VITAL trial provides further evidence for assessing the effects of vitamin D and *n*-3 PUFAs on the primary prevention of CVD and cancer. Although there was no direct link between *n*-3 PUFAs and major composite cardiovascular events, significant findings were observed across subgroups for both total and fatal MI [69]. In the STRENGTH trial, there was no cardiovascular benefit in patients treated with *n*-3 PUFAs in the free acid form. However, this trial only evaluated patients with a high risk of developing cardiovascular disease, and the effects of *n*-3 PUFAs may differ in lower-risk primary prevention populations who have yet to develop significant atherosclerosis or clinical conditions [14].

The effectiveness of *n*-3 PUFAs in treating dyslipidemia, particularly hypertriglyceridemia, is widely accepted [12,70,71,72], and this effect may be attributed to *n*-3 PUFA’s ability to accelerate the breakdown of chylomicron particles and reduce the production of very low-density lipoprotein (VLDL) cholesterol [73]. In addition to their effects on lipids, animal experiments and clinical intervention studies have also shown that *n*-3 PUFA possesses anti-inflammatory and antithrombotic effects [74,75,76]. Hence, the multiple physiologic advantages of *n*-3 PUFA may assist patients with atherosclerotic arteries [77].

ALA and DPA, unlike EPA and DHA, have received less attention. Our results are similar to those reported in previously published meta-analyses, indicating that higher ALA exposure is associated with a lower CHD risk [24]. The Singapore Chinese Health Study enrolled 63,257 adults and reported an HR of 0.81 (0.72–0.90) for the association between higher ALA intake and the risk of cardiovascular mortality [78]. However, in another large prospective cohort study from the Netherlands, ALA was only negatively linked with stroke, not the clinical endpoint of CHD [79]. However, ALA is susceptible to beta-oxidation, and a small portion is converted to EPA. Thus, ALA concentrations might not accurately reflect the ALA consumption [80].

In earlier investigations, the outcomes related to *n*-3 DPA have been inconsistent [30,38,81]. Mozaffarian and colleagues [38] conducted a sizeable cohort and found that higher plasma DPA levels were associated with lower all-cause mortality. However, Sun et al.’s prospective analysis demonstrated that lower levels of DPA were connected with a higher risk of non-fatal MI [30].

The concentration of *n*-3 PUFAs in blood and tissues seems to be essential for the pathophysiological effects in humans, yet few meta-analyses have compared the levels of *n*-3 PUFAs in patients with and without CHD. One concern about retrospective studies is that individuals with the disease of interest may have altered their diets in response to the diagnosis, or the disease process itself may have altered biomarker levels. This appears not to have been a concern in the present study since levels of *n*-3 PUFAs (EPA, DPA, DHA) were lower, not higher, in the CHD patients than in the controls. A case-control study by Block et al. suggested that there may be an inverse relationship between the levels of *n*-3 PUFAs and the risk of acute coronary syndrome (ACS), indicating that blood EPA + DHA levels may contribute to risk stratification [46]. Since the cardioprotective effects of omega-3 have been touted for well over 30 years now, patients with CHD might have been expected to have increased their intake of EPA and DHA, in which case the CHD patients might have had higher levels than controls. The finding that levels are lower in the cases argues against this concern and suggests that reverse causation is likely not playing a role in these studies.

In our subgroup analysis, EPA + DHA concentrations were negatively associated with total CHD, fatal and non-fatal CHD. For the different *n*-3 PUFA exposure compartments, the *n*-3 PUFA levels in erythrocytes were more closely correlated with CHD than for whole blood, plasma, and serum. *n*-3 PUFAs are typically found in cell membranes (to a small extent in adipose tissue), and RBCs have many qualities that make them a good sample type for monitoring *n*-3 PUFA status [82]. Compared to erythrocytes, serum, plasma, and whole blood may be affected by more factors. When comparing the utility of different biological matrices as biomarkers for CHD risk, erythrocytes may offer certain advantages over serum, plasma, and whole blood. Erythrocytes, by contrast, may provide a more stable environment for the measurement of certain biomarkers due to their relatively consistent lifecycle and composition [83].

The current study has several advantages: (1) We assessed different types of *n*-3 PUFA biomarkers separately. Furthermore, we performed subgroup analyses of different clinical endpoints (total CHD, fatal and non-fatal CHD) as well as different exposure compartments. (2) In contrast to the self-report questionnaires used in many studies, we limited our study selection to those that show measurements of *n*-3 PUFA in blood or tissue, which reduced the recall bias and improved the accuracy of the results. (3) By including studies from 13 countries, we minimized publication bias and increased generalizability. Sensitivity analyses showed our results were robust. This study also has some important limitations. More trials and detailed subgroup analysis information are needed to fully evaluate true subgroup effects. Furthermore, considerable heterogeneity exists between studies, which is hard to avoid due to differing methods for measuring PUFAs and pooling data from multiple sample types that can influence results. Dietary conditions in various geographic regions may also contribute to this study’s heterogeneity. Lastly, for the prospective studies, blood samples were analyzed only once at baseline. Theoretically, levels could have changed during the follow-up period, particularly if participants who were not taking supplements at baseline began taking them later.

## 5. Conclusions

Higher levels of circulating *n*-3 PUFAs (ALA, EPA, DPA, DHA, EPA + DHA) are associated with a reduced risk of CHD. Furthermore, CHD patients had significantly lower levels of *n*-3 PUFAs (except ALA). Our findings support a protective role of *n*-3 PUFAs on CHD events.

## Figures and Tables

**Figure 1 nutrients-16-01610-f001:**
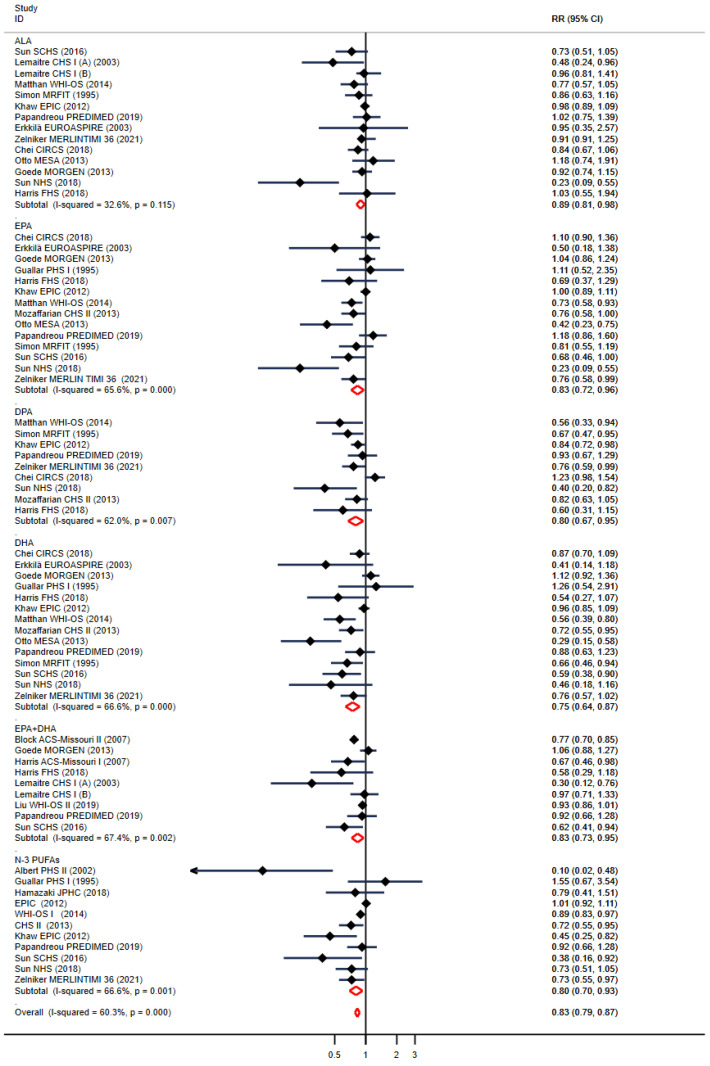
Pooled relative risk between individual *n*-3 PUFA levels and incident coronary heart disease (CHD). RRs for individual trials; horizontal lines indicate 95% confidence interval (CI); ALA: α-linolenic acid; DPA docosapentaenoic acid; EPA: eicosapentaenoic acid; DHA docosahexaenoic acid; *n*-3 PUFA: *n*-3 polyunsaturated fatty acid.

**Figure 2 nutrients-16-01610-f002:**
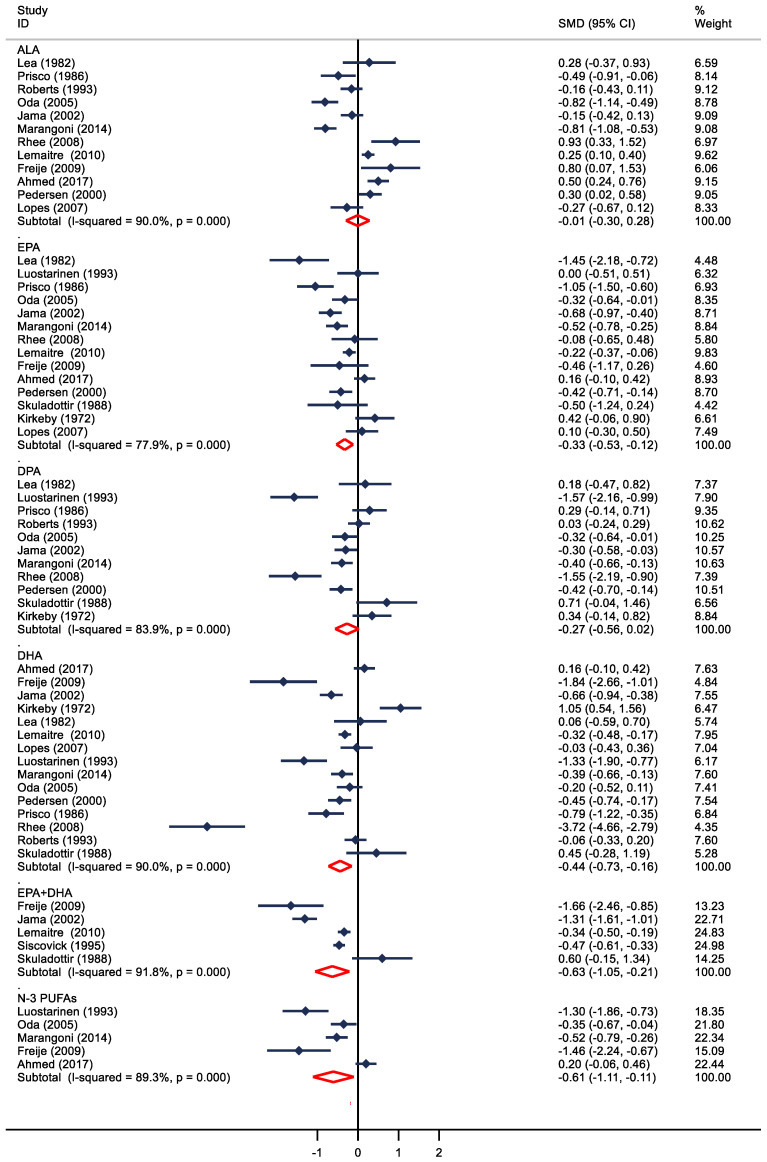
Differences between individual *n*-3 PUFA levels in patients with and without coronary heart disease (CHD). A negative point estimate means that levels were lower in CHD cases vs. controls. Weight mean differences (WMDs) for individual trials; horizontal lines indicate 95% confidence interval (CI); ALA: α-linolenic acid; DPA docosapentaenoic acid; EPA eicosapentaenoic acid; DHA docosahexaenoic acid; *n*-3 PUFA: *n*-3 polyunsaturated fatty acid.

**Figure 3 nutrients-16-01610-f003:**
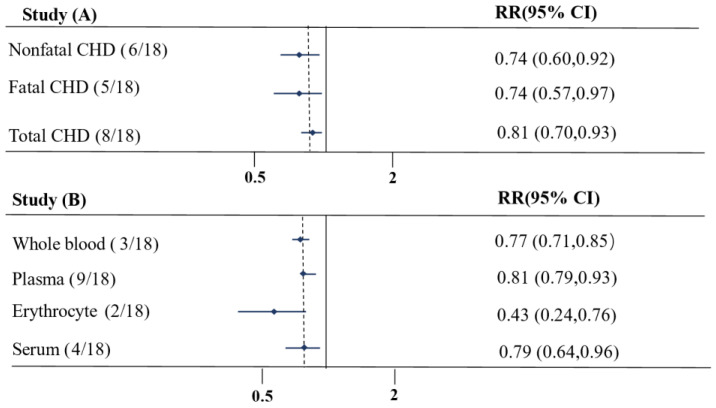
Pooled relative risk subgroup analysis between *n*-3 PUFA levels and incident coronary heart disease (CHD). RRs for the summary estimate of the subgroup analysis (Appendix A); horizontal lines indicate 95% confidence interval (CI); CHD: coronary heart disease; (**A**) clinical endpoint; (**B**) lipid pools.

**Table 1 nutrients-16-01610-t001:** Baseline characteristics of prospective studies in the pooled analysis.

Cohort	Total N	Study	Year	Country	N	SEX (Male%)	Design	Age, y	Follow-Up, y	Lipid Pool	Outcomes
SCHS	63,257	Sun	2016	Singapore	744/744	64.79	PC, NCC	66.1	11	Plasma	AMI
CHS I	5201	Lemaitre	2003	USA	179/179	57.4	PC, NCC	79.1	7	Plasma	MI
WHI-OS I	93,676	Matthan	2014	USA	1224/1224	0	PC, NCC	67.8	4.5	Plasma	CHD
MRFIT	/	Simon	1995	USA	94/94	100%	PC, NCC	49.8	3	Serum	CHD
EPIC	25,639	Khaw	2012	UK	2424/4930	81	PC, NCC	64.9	4	Plasma	CHD
PREDIMED	7447	Papandreou	2019	Spain	136/272	61.3	PC, NCC	67.8	7	Whole blood	CHD
EUROASPIRE	/	Erkkilä	2003	Finland	334/493	80	PC, NCC	59	3	Serum	AMI
MERLIN TIMI 36	/	Zelniker	2021	USA	528/1612	73.8	PC	66.3	/	Serum	CHD
CIRCS	12,840	Chei	2018	Japan	152/456	61	PC, NCC	/	8	Serum	CHD
MESA	/	Otto	2013	USA	736/2837	46.8	PC, NCC	61.5	11	Plasma	CHD
MORGEN	35,475	Goede	2013	Netherlands	279/279	70	PC, NCC	50.5	8	Plasma	Fatal CHD
PHS I	14916	Guallar	1995	USA	213/213	100	PC, NCC	58.7	5	Plasma	MI
NHS	32,826	Sun	2008	USA	146/288	/	PC, NCC	60.3	6	Erythrocyte	Non-fatal MI
CHS II	30,829	Mozaffarian	2013	USA	630/2692	36.3	PC	74	/	Plasma	Fatal CHD
FHS	2500	Harris	2018	USA	119/2500	43	PC	66	7.3	Erythrocyte	CHD
ACS-Missouri I	/	Harris	2007	USA	94/94	54.3	PC, NCC	46.4	/	Whole blood	ACS
ACS-Missouri II	/	Block	2008	USA	768/768	66	PC, NCC	61	/	Whole blood	ACS, MI
PHS II	22,071	Albert	2002	USA	94/184	/	PC, NCC	58.5	17	Whole blood	SCA
JPHC	116,896	Hamazaki	2018	Japan	209/418	63.6	PC, NCC	57.1	16	Plasma	CHD
WHI-OS II	93,676	Liu	2019	USA	1214/1214	0	PC, NCC	67.8	/	Plasma	CHD

SCHS: Singapore Chinese Health Study; CHS: Cardiovascular Health Study; WHI-OS: Women’s Health Initiative observational study; MRFIT: Multiple Risk Factor Intervention Trial; EPIC: European Prospective Investigation into Cancer; PREDIMED trial: Prevención con Dieta Mediterránea; EUROASPIRE study: European Action on Secondary Prevention through Intervention to Reduce Events; ERLINTIMI 36:Metabolic Efficiency With Ranolazine for Less Ischemia in Non–ST-Elevation Acute Coronary Syndrome Thrombolysis in Myocardial Infarction (36); CIRCS: Circulatory Risk in Communities Study; MESA: Multi-Ethnic Study of Atherosclerosis; MORGEN (MP-2): Monitoring Project on Cardiovascular Disease Risk Factors and Chronic Diseases; PHS: Physicians’ Health Study; NHS: Nurses’ Health Study; CHS: Cardiovascular Health Study; FHS: Framingham Heart Study; ACS study: acute coronary syndrome study; JPHC: The Japan Public Health Center-based study; AMI: acute myocardial infarction; ACS: acute coronary syndromes; SCA: sudden cardiac arrest; PC: prospective cohort study; NCC: nested case-control study; Total CHD events were defined as non-fatal CHD (angina, non-fatal myocardial infarction (MI), and fatal CHD (fatal MI and sudden cardiac death); y: years.

**Table 2 nutrients-16-01610-t002:** Baseline characteristics of case-control studies in the pooled analysis.

Study	Year	Country	N	SEX (M)%	Design	Age, y	Biomarker	Disease
Lea	1982	Britain	20/17	/	CC	/	Erythrocyte	MI
Luostarinen	1993	Sweden	30/29	/	CC	40	Tissue	SCD
Prisco	1986	Italy	42/45	57	CC	51.2	Erythrocyte	CHD
Roberts	1993	USA	66/292	100	CC	25–64	Adipose tissue	SCD
Oda	2005	Japan	73/84	84	CC	65	Serum	AMI
Jama	2002	Norway	103/104	71.9	CC	62.7	Serum	MI
Marangoni	2014	Italy	119/103	/	CC	55.9	Whole blood	MI
Rhee	2008	Korea	30/20	0	CC	35.5	Plasma	CHD
Lemaitre	2009	USA	265/415	81.3	CC	58.4	Erythrocyte	SCD
Freije	2009	Bahrain	11/26	45.5	CC	18–57	Erythrocyte	CHD
Ahmed	2017	Tunisia	111/120	58	CC	60.8	Plasma	CAD
Pedersen	2000	Norway	100/98	72	CC	62.4	Adipose tissue	MI
Skuladottir	1988	Iceland	12/14	/	CC	66.3	Serum	CHD, Fatal MI
Kirkeby	1972	Norway	36/32	100	CC	59.1	Serum	MI
Siscovick	1995	USA	82/108	80	CC	59	Erythrocyte	SCD
Lopes	2007	Portugal	49/49	100	CC	56.6	Adipose tissue	MI

CC: case-control; SCD: sudden cardiac death; y: year.

## Data Availability

The raw data supporting the conclusions of this article will be made available by the authors on request.

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
