# Peer review of "Circulating Omega-3 Polyunsaturated Fatty Acids Levels in Coronary Heart Disease: Pooled Analysis of 36 Observational Studies"

_nutrients, 2024, doi:10.3390/nu16111610_

Round 1

Reviewer 1 Report

Comments and Suggestions for Authors

This is a meta-analysis of published papers reporting omega-3 levels and CHD outcomes.

33. Simply say “CHD, with RRs (95% CI) of 0.89 (0.82,0.98)…” etc. No need to say 95% CI every time.

38. Stronger than what??

48. These FAs were not ‘discovered’ in the 1970s; they were known back in the ‘50s. What happened in the late 1970s was that they were brought to the attention of the medical community by Dyerberg and Bang.

52. I would move ‘anti-thrombosis’ up with lipids and hypertension since the anti-platelet effects are not actually driven by the SPM metabolites – prostaglandins are oxylipins but they are not SPMs.

54. I think you mean “As recommended by the AHA Dietary Guidelines since 2003, the health benefits of n-3 PUFAs for both primary and secondary prevention of cardiovascular disease can be achieved by eating at least two servings of fish per week.”

60. remove “pooled”

61ff. You may want to add Chao (1), Rizos(2) , and Bernasconi (3) to your list of recent meta-analyses.

72. caused instead of affected. Also, “One factor to consider is the inconsistency in fatty acid mixing [what is this?], distribution, and metabolism across different populations.” This sentence is unclear and does not really say anything of substance. Please delete or elaborate.

75. Ref 19 has nothing to do with medications; it’s about FADS genotypes. If there is an effect of medications, please explain – I’m not aware of any except perhaps fat absorption blockers.

76. “self-reported assessments”… what is this? Self-reported heart attacks or strokes? Most studies rely on medical records. Perhaps you mean self-reported dietary intake of omega-3 or fish. Agree with that; provide a citation.

84. Chao et al noted earlier looked at om3 RCTs in CAD patients… but did not compare cases to controls per se.

85. All of the studies here are observational; the essential distinction is between retrospective and prospective.

102. This is better titled Exposures and Outcomes

103. Assessment of fatty acid… what? Levels? Classes?

121. Better called ‘lipid pools’ or ‘sample types’… and the various n3 PUFAs are not actually biomarkers – they are fatty acids. The term ‘fatty acid biomarkers’ is redundant.

142. Those are not “intervals”… those are lipid pools.

143. Those are more CHD ‘outcomes’ than diagnoses

Tables 1 and 2.  Obviously these need better formatting since some text-wrapping is occurring. It would be helpful to the reader to get some summary statistics such as total N (not sample size), sex % MW, mean age and mean yrs of follow-up. It’s not clear what “disease” means… is that the outcome reported and if so, is that the only one? I would call this column “Outcomes” Also again, serum is not a ‘biomarker’… it’s a lipid pool or sample type.

Figure 1 title. Maybe “and incident CHD” is better

Figure 2 title. “differences between individual n-3 PUFA levels in patients with and without CAD.” Make it clear that a negative point estimate means that levels were lower in cases vs controls. Maybe put “lower in patients” and “higher in patients” under the X-axis on either side of 0.

Figure 3 does not appear to have a legend. And “Tatal” probably should be Total?

229. It’s not true that there was a better correlation for blood, serum and plasma (by the way, there is no need whatever to separate serum and plasma – for fatty acids, they are the same thing) than for RBCs. Actually the point estimate was stronger for RBCs than the other sample types, it’s just that the variability was greater for RBCs than the other metrics. It would be important to report the EPA+DHA values for cases and controls, and for Qlow vs Qhigh (whatever you used) for these studies in supplementary data. I’m wondering if the RBC data were very different between studies.

Supplementary data. I don’t understand what Supplementary Table T2 means. There should be some explanation accompanying.

238. Better said, “The present study was a comprehensive…”

249. “benefits” has a therapeutic ring to it, like in an RCT. I’d say ‘but there were exceptions [15].”

 253. “[9], where the risk of …”

255. “However, the outcomes of other RCTs using only 840 mg of EPA+DHA were less favorable. ORIGIN and ASCEND found that…”

261. treated with n-3 PUFAs in the free acid form.”

282. Should cite a paper for Mozaffarian

289. I think you could add something like the following in this section. “One concern about retrospective studies is that individuals with the disease of interest may have altered their diets in response to the diagnosis, or the disease process itself may have altered biomarker levels. This appears not to have been a concern in the present study since levels were lower, not higher, in the CHD patients than the controls. Since the cardioprotective effects of omega-3 have been touted for well over 30 years now, patients with CHD might have been expected to have increased their intake of EPA and DHA, in which case the CHD patients might have had higher levels than controls. The finding that levels are lower in the cases argues against this concern and suggests that reverse causation is likely not playing a role in these studies.”

299. FA “conversion” does not take place in membranes. If you mean the “conversion” of ALA into EPA and DHA, that takes place in the liver. What is true is that n-3 PUFAs are typically found in cell membranes (and to a small extent in adipose tissue), and that RBCs have many qualities that would make them a good sample type for monitoring omega-3 status. You might cite Harris 2008.(4)

306-07. not a complete sentence, but it appears to be referring to reference 29 by Harris. In that paper, there are no CHD outcomes reported … only total mortality. It’s unclear why it is included in this analysis.

312. Well, after that last comment, I wonder about your first advantage!

318. Including studies from many countries has nothing to do with publication bias, but it does, as you say, improves generalizability

I would add as limitations the fact that for the prospective studies, blood was only analyzed once at baseline, and theoretically, levels could have changed during follow up, especially if people were not taking supplements at baseline but began taking them later.

1.           Chao T, Sun J, Ge Y, Wang C. Effect of omega-3 fatty acids supplementation on the prognosis of coronary artery disease: A meta-analysis of randomized controlled trials. Nutrition, metabolism, and cardiovascular diseases : NMCD 2024;34(3):537-47.

2.           Rizos EC, Markozannes G, Tsapas A, Mantzoros CS, Ntzani EE. Omega-3 supplementation and cardiovascular disease: formulation-based systematic review and meta-analysis with trial sequential analysis. Heart 2021;107(2):150-8.

3.           Bernasconi AA, Lavie CJ, Milani RV, Laukkanen JA. Omega-3 Benefits Remain Strong Post-STRENGTH. Mayo Clinic proceedings 2021.

4.           Harris WS. The omega-3 index as a risk factor for coronary heart disease. Am J Clin Nutr 2008;87:1997S-2002S.

Comments on the Quality of English Language

Pretty good

Author Response

Thank you for sharing your opinion. Your insights are incredibly professional, detailed, and insightful, clearly reflecting a deep expertise in the field of FAs. Your comments have significantly enhanced the quality of our paper, enriching not only the content but also broadening my understanding of the topics discussed. This deeper perspective is invaluable. I am grateful for the time and effort you put into your review. Please let us know if you have any other questions or require any further information.

Please see the attachment for details.

Reviewer 2 Report

Comments and Suggestions for Authors

The authors present an evaluation of the association between omega-3 levels and incidence of CV events. The work may offer some interesting points, although the topic is not particularly innovative.

TITLE: The title should be changed: this is not an individual-level meta-analysis because the analysis is not based on data from individual patients.

OBJECTIVES: What do the authors mean by 'strictly distinguished between observational and retrospective studies'? The retrospective studies included are still observational. Furthermore, it is wrong to consider grafted case-control studies as prospective.

Since measured and not self-reported values are used, why is this distinction relevant?

METHODS AND RESULTS: The assessment of the quality of the studies is missing.

TABLE1: please, use 'Country' instead of 'Race'

DISCUSSION: The discussion should begin by comparing the data from this meta-analysis with other studies that have evaluated the same association, i.e. omega-3 levels and incidence of events. Studies that have evaluated clinical outcomes from therapeutic supplementation are conceptually different and should be discussed next.

Comments on the Quality of English Language

Minor editing of English language required

Author Response

We greatly appreciate you taking the time to provide such comprehensive and thoughtful feedback. We believe the revisions and additions in response to your feedback have substantially improved the manuscript. Please let us know if you have any other questions or require any further information.

Please see the attachment for details.

Reviewer 3 Report

Comments and Suggestions for Authors

This is a comprehensive well presented analysis of an important but complex topic. The strength is the focus on objectively measured N-3-PUFA data and the comprehensive discussion.

Some further specific discussion on the benefits of EPA vs DHA  should be included in the second paragraph of the discussion given the recent outcome trials that are mentioned there eg REDUCE -IT See also how you present figure 3 and the related text.

There are some problems with the tables, the table and figure legends and the text below them. 

Comments on the Quality of English Language

Can do with some improvements

Abstract:

Lines 37 and 38 - Should it read was generally stronger than for erthrocytes.

Introduction:

line 4 Should it be reported rather than discovered

Lines 54-56 Should be to supplement and then the whole sentence needs to be rewritten suggest - As recommended ....2003, there are health benefits from eating...

Line 83 - Do you mean no pooled studies?

Methods 2.2 Line 104

Suggest changing "particpating studies" to "the studies included in our analysis"

Methods 2.5

line 142 I dont know what you mean by interval.  Do you mean duration?

Tables 1 and 2

There are aligment problems eg Singapore should be on one line.

Clarify N especially when you have two components eg 334/493

Below table 2 

Its hard to tell what's figure legend and what is the main text - line 188 onwards

Similar issues for the other tables and figures

Figure 3 needs a headin. Also what is A and what is B in figure 3?

Author Response

Thank you very much for the time and effort you spent reviewing my manuscript. Your detailed analysis and constructive criticism have greatly helped in refining my arguments and strengthening the data presentation. Thank you once again for your rigorous review and insightful comments.

Please see the attachment for details.

Reviewer 4 Report

Comments and Suggestions for Authors

The paper is interesting, but there are some points which deserve adjustment.

 Title: “individual-level pooled analysis “– from the “Data extraction” section it does not appear that individual data are used.  IPD meta-analyses, rather than extracting summary (aggregate) data from study publications, aim to obtain the original research data directly from the researchers responsible for each study. The present paper does not appear to be IPD meta-analysis.

 Line 104 - the abbreviation FA (Fatty Acid, I suppose) is not defined.

Line 111 – in nested case-control studies hazard ratio is not the statistic we expect to find. If so, why is it mentioned here? If any study used this statistic, how was it converted to risk ratio?

Line 138 – if a random-effect model was applied when I squared was greater than 50%, is the analysis represented in figure 1 analyzed partly with fixed-effect model and partly with random-effect model? Keeping into account the low statistical power of I squared, probably using always a random-effect model would be sensible; it in fact assumes that the differences in the results drawn from each study are either random or due to differences between the populations studied or related to the characteristics of the individual studies. In any case, the statement contradicts line 26.

Line 190 – the Authors state the division of the CHD events in non-fatal and fatal. Which events are used as endpoint in the forest plot in figure 1?

Line 190-196 – repeating in the text all the numbers presented in figure 1 seems redundant. The same is true for lines 209-212.

Line 231 – figure 3 has no legend. It should make clear that the reported RR is the summary estimate of the subgroup analysis reported in Supplement.

Table 1 – the length of the follow-up is widely heterogeneous (the unit of measurement is not reported, I suppose years). Why a subgroup analysis according to follow-up length was not performed?

Table S1 – Supplement – almost all the items in the first paragraph of the Pubmed search strategy are included in the Fatty Acids, Omega-3 MeSH term, so the emphasis (line 96) on inclusion of synonyms is not justified.

Enclosed with the manuscript was a file with the Arrive Essential 10. The meaning is not clear, as no animal study is included in the review.

Comments on the Quality of English Language

The quality of English language is adequate, and it deserves only minor editing

Author Response

Thank you very much for the time and effort you spent reviewing my manuscript. Highly professional in this field. Your detailed analysis and constructive criticism have greatly helped in refining my arguments and strengthening the data presentation. Thank you once again for your rigorous review and insightful comments. 

Please see the attachment for details.

Round 2

Reviewer 1 Report

Comments and Suggestions for Authors

The authors have responded adequately to my concerns

Author Response

Thank you again for your valuable input.

Reviewer 2 Report

Comments and Suggestions for Authors

The authors have modified the article according to the requests. However, some key points remain to be addressed:

MAJOR

The quality of the included studies must be assessed even if they are observational studies. This is not related to publication bias, but to the robustness and validity of the methods and study design. There are various instruments to assess the quality of observational studies. For example, the Newcastle-Ottawa scale can be applied.

The analysis presented concerns the association between omega-3 levels and cardiovascular outcomes. This association is different from that between omega 3 supplementation and cardiovascular outcomes. Although the two pieces of evidence may support each other, it is not obvious that one implies the other. It is important to start discussion comparing studies that have tested the association between levels (not necessarily modified by nutritional or pharmacological interventions) and CV outcomes. See, for example, Zhang Y, et al Elife. 2024 Apr 05;12

MINOR

RESULTS: 'The characteristics of the 20 prospective cohorts and 16 retrospective case-control trials are presented', please change with 'The characteristics of the 20 prospective cohorts and 16 retrospective case-control studies are presented'

TABLE 2: also here, please, use 'Country' instead of 'Race'

Author Response

Thank you for your professional opinion. Your insights have been very helpful. I have made several revisions and updates to the manuscript.

Please see the attachment for details.

Reviewer 4 Report

Comments and Suggestions for Authors

The paper is interesting, and worth publishing, but there are some points which deserve adjustment.

Title: “individual-level pooled analysis “– from the “Data extraction” section it does not appear that individual data are used.  IPD meta-analyses, rather than extracting summary (aggregate) data from study publications, aim to obtain the original research data directly from the researchers responsible for each study. The present paper does not appear to be IPD meta-analysis.

Line 104 - the abbreviation FA (Fatty Acid, I suppose) is not defined.

Line 111 – in nested case-control studies hazard ratio is not the statistic we expect to find. If so, why is it mentioned here? If any study used this statistic, how was it converted to risk ratio?

Line 138 – if a random-effect model was applied when I squared was greater than 50%, is the analysis represented in figure 1 analyzed partly with fixed-effect model and partly with random-effect model? Keeping into account the low statistical power of I squared, probably using always a random-effect model would be sensible; it in fact assumes that the differences in the results drawn from each study are either random or due to differences between the populations studied or related to the characteristics of the individual studies. In any case, the statement contradicts line 26.

Line 190 – the Authors state the division of the CHD events in non-fatal and fatal. Which events are used as endpoint in the forest plot in figure 1?

Line 190-196 – repeating in the text all the numbers presented in figure 1 seems redundant. The same is true for lines 209-212.

Line 231 – figure 3 has no legend. It should make clear that the reported RR is the summary estimate of the subgroup analysis reported in Supplement.

Table 1 – the length of the follow-up is widely heterogeneous (the unit of measurement is not reported, I suppose years). Why a subgroup analysis according to follow-up length was not performed?

Table S1 – Supplement – almost all the items in the first paragraph of the Pubmed search strategy are included in the Fatty Acids, Omega-3 MeSH term, so the emphasis (line 96) on inclusion of synonyms is not justified.

Enclosed with the manuscript was a file with the Arrive Essential 10. The meaning is not clear, as no animal study is included in the review.

Author Response

Thank you again for your valuable input. Your insights have been very helpful. I have made several revisions and updates to the manuscript.

Please see the attachment for details.
